# Feasibility of Differential Dose—Volume Histogram Features in Multivariate Prediction Model for Radiation Pneumonitis Occurrence

**DOI:** 10.3390/diagnostics12061354

**Published:** 2022-05-31

**Authors:** Yoshiyuki Katsuta, Noriyuki Kadoya, Yuto Sugai, Yu Katagiri, Takaya Yamamoto, Kazuya Takeda, Shohei Tanaka, Keiichi Jingu

**Affiliations:** 1Department of Radiation Oncology, Tohoku University Graduate School of Medicine, 1-1 Seiryo-machi, Aoba-ku, Sendai 980-8574, Japan; kadoya.n@rad.med.tohoku.ac.jp (N.K.); sugai.y@med.tohoku.ac.jp (Y.S.); priere_for@yahoo.co.jp (T.Y.); takeda7616@gmail.com (K.T.); s1290169@gmail.com (S.T.); kjingu-jr@rad.med.tohoku.ac.jp (K.J.); 2Department of Radiation Oncology, Japan Red Cross Ishinomaki Hospital, Ishinomaki 986-8522, Japan; ykhasedo@yahoo.co.jp

**Keywords:** radiotherapy, NSCLC, radiation pneumonitis, machine learning, artificial intelligence

## Abstract

The purpose of this study is to introduce differential dose–volume histogram (dDVH) features into machine learning for radiation pneumonitis (RP) prediction and to demonstrate the predictive performance of the developed model based on integrated cumulative dose–volume histogram (cDVH) and dDVH features. **Materials and methods:** cDVH and dDVH features were calculated for 153 patients treated for non-small-cell lung cancer with 60–66 Gy and dose bins ranging from 2 to 8 Gy in 2 Gy increments. RP prediction models were developed with the least absolute shrinkage and selection operator (LASSO) through fivefold cross-validation. **Results:** Among the 152 patients in the patient cohort, 41 presented ≥grade 2 RP. The interdependencies between cDVH features evaluated by Spearman’s correlation were significantly resolved by the inclusion of dDVH features. The average area under curve for the RP prediction model using cDVH and dDVH model was 0.73, which was higher than the average area under curve using cDVH model for 0.62 with statistically significance (*p* < 0.01). An analysis using the entire set of regression coefficients determined by LASSO demonstrated that dDVH features represented four of the top five frequently selected features in the model fitting, regardless of dose bin. **Conclusions:** We successfully developed an RP prediction model that integrated cDVH and dDVH features. The best RP prediction model was achieved using dDVH (dose bin = 4 Gy) features in the machine learning process.

## 1. Introduction

In radiotherapy, care for non-small-cell lung cancer (NSCLC) is challenging because healthy lung tissues as well as numerous organs, including the esophagus, trachea, and brachial plexus, are located near the tumor [1]. The adverse effects of this treatment often include radiation pneumonitis (RP), which is fatal in some cases [2]. Thus, the accurate prediction of severe RP is essential for the efficacious and safe treatment of NSCLC [3]. Previous studies have provided various features for RP prediction, such as V20Gy (the fractional volume of healthy lung tissues receiving doses exceeding 20 Gy) and mean lung dose (MLD) [1,2,3]. However, even patients treated with simultaneous use of these features and modern irradiation techniques that achieve planning target volume coverage and organs at risk sparing, such as volumetric modulated arc therapy, experience serious side effects in some cases [4]. Therefore, the interest and challenges in the prediction of RP remain as high as ever.

As one of the next-generation strategies, a number of researchers have already proposed the application of machine learning to RP prediction [5,6,7]. Machine learning, which offers a data-driven approach to the development of RP prediction models, presents the rules among many features and predicts RP occurrence based on the integration of these features. Integrations of dosimetric features for RP prediction models using the least absolute shrinkage and selection operator (LASSO) [5], random forests [6], and support vector machines [7] have demonstrated effective RP prediction.

One of the common problems in the development of RP prediction models is that the cumulative dose–volume histogram (cDVH) features often integrated into the models [5,6,7] supply only a few points from the dose–volume curve, potentially leaving out helpful information that could be used to discriminate between treatment plans introducing RP occurrence [8]. Additionally, such models often include multicollinearity [9], that is, a dependency between multiple explanatory variables or near-linear dependencies between features. In fact, introducing these interdependencies into machine learning processes often prevents the development of an optimal prediction model [10]. cDVH features commonly used to develop prediction models for radiotherapy partly include common volumes of normal tissues, which end up receiving larger than the specified dose. For example, for V5Gy and V20Gy, volumes with >20 Gy will be included for both of these cDVH features. These dependencies between the explanatory variables obscure the effect of each explanatory variable on the objective variable, resulting in an unreliable prediction model. In fact, there is no guarantee of improvement due to the addition of features, since the introduction of redundant information can make the model dangerously prone to overfitting [11]. In contrast, the use of differential dose–volume histogram (dDVH) features calculated from dose–volume curves, which do not include the common volume of normal tissues, has been explored by several researchers [8,12], especially in conjunction with cDVH features. In this study, we for the first time introduce dDVH features into the development of an RP prediction model using machine learning and demonstrate the model’s predictive performance for RP occurrence.

## 2. Materials and Methods

### 2.1. Patient Cohort

Our patient cohort included 153 patients with NSCLC treated with 3D conformal radiation therapy (3DCRT) from 2006 to 2019 at Tohoku University Hospital. This study was approved by the ethical review board at our hospital (2020-1-1066). Patients received 2.0 Gy fractions for a total of 60–66 Gy using a 6- or 10-MV external photon beam and required a 6-month follow-up period at minimum.

### 2.2. Radiation Treatment Planning and Dose–Volume Histogram Features

Patients in our cohort were scanned for treatment planning using Bright Speed 16 (GE Healthcare, Milwaukee, WI, USA) or SOMATOM Definition Flash AS (Siemens Healthcare, Erlangen, Germany) with a pixel size of <1.0 mm and a slice thickness of <2.5 mm.

We used either an analytical anisotropic algorithm or an Acuros XB dose calculation algorithm equipped in an Eclipse treatment planning system (TPS; Varian Medical Systems, Palo Alto, CA, USA) with a calculation grid of 2 mm. The number of patients treated with the Varian CL2100, CLINAC iX, and TrueBeam STx were 41, 109, and 3, respectively. The dose to normal lung was constrained within V5Gy < 65%, V20Gy < 35%, and MLD < 20 Gy, considering the RTOG0617 protocol [13] and while maintaining an adequate target volume coverage at the time the treatment plans were created.

Both cDVH and dDVH were computed from a dose–volume curve calculated using CT images, a structure set, and calculated dose exported from the Eclipse TPS in-house using MATLAB (MathWorks, Natick, MA, USA). cDVH features, including cV_5Gy_ and cV_10Gy_–cV_60Gy_ (in 10-Gy increments), and MLD, were also computed. Each dDVH feature was calculated between 5 and 60 Gy with dose bins of 2–8 Gy in 2-Gy increments (resulting in four patterns).

### 2.3. Prediction Model Development

Prediction models were developed using LASSO logistic regression after feature conversion to Z scores (mean = 0, standard deviation = 1), as recommended by Kang et al. [14]. The following LASSO equation (Equation (1)) achieves the development of a predictive model through simultaneous execution of coefficient determination and feature selection by penalizing the L1 norm:(1)$$\mathrm{LASSO}=\underset{w}{\min}\overset{m}{\underset{i=1}{\sum }}{\left(y_{i}-w^{T}x_{i}\right)}^{2}+\lambda \|w\|$$

The parameter *λ* is the penalty strength, which determines the magnitude of the coefficients and eliminates redundant features. LASSO shrinks the associated coefficients to zero as the penalty is strengthened, resulting in a more regularized model. The model developed by LASSO shows the relationship between RP occurrence and one or more risk factors with coefficients representing every risk factor.

Our prediction model was developed via six iterations of fivefold double cross-validation, with the total data set being split into training, validation, and testing data sets. For the inner loop, the training set and validation set were used to optimize the penalty tuning parameter *λ* based on fivefold cross-validation. The most regularized model was determined by searching for the *λ* that minimized the mean loss at cross-validation. The prediction performance that led to the optimal *λ* was obtained in the outer loop using the testing data set. A testing data set independent from the training and validation data sets was also used in the inner loop for model optimization. Four sets of candidate models were used to determine the capability of dDVH features in the development of RP prediction models: (1) cDVH features and (2) +dDVH features (resulting in four patterns).

An evaluation of prediction performance was conducted for each prediction model using the area under curve (AUC) and an extraction of the coefficients for each feature. In total, 70% of the data were used in the inner loop (sum of the training and validation data sets), and 30% were used in the outer loop (testing data set), considering the overall data set size. *p*-values were computed from paired Wilcoxon signed-rank test.

## 3. Results

As summarized in Table 1, 118 patients (78.4%) received 60 Gy. At the same time, 41 patients (26.8%) had ≥grade 2 RP based on the Common Terminology Criteria for Adverse Events (version 4.0). As shown in Figure 1, smaller dose points on the dose–volume curve indicated larger cDVH features. In our patient cohort, the mean ± SD of 24.0% (7.5%) for cV_20Gy_ increased to 30.2% (9.1%) for cV_10Gy_ and to 38.5% (11.1%) for cV_5Gy_. In contrast, the dDVH features were inhomogeneously distributed, as shown in Figure 2, revealing that these features captured the doses to lung tissues in more detail compared with cDVH features. For instance, the dDVH (bin = 4 Gy) features (B) revealed that the doses of 37–41 Gy and 41–45 Gy (dV_37–41Gy_ and dV_41–45Gy_) received by the lungs were larger than the adjacent doses of 33–37 Gy and 45–49 Gy (dV_33–37Gy_ and dV_45–49Gy_). Moreover, for every bin width, dDVH features calculated the lowest dose ranges (dV_5–7Gy_, dV_5–9Gy_, dV_5–11Gy_, and dV_5–13Gy_) based on the distance between them.

Figure 3 shows the dose distribution, cDVH curve, and dDVH features of representative examples of ≥grade 2 RP. For ≥grade 2 RP patients, changes in dV_5Gy_ value relative to the dV_20Gy_ value of 4.1% were larger than those in cV_5Gy_ value relative to the cV_20Gy_ value of 1.4%. Similar results were obtained in case of <grade 2 RP (3.6% vs. 1.7%). These results indicate that dDVH features express numerical dose distribution more clearly than do cDVH features, regardless of RP grade.

Figure 4 shows a heat map depicting Spearman’s correlation, which was applied for the observation of interdependencies [15,16,17] between cDVH features. The correlations between cDVH features (A), computed from points near to each other on the dose–volume curve, were large, and they decreased as distance between points increases. For cV_5Gy_, the correlation of 0.97 at cV_10Gy_ decreased to 0.33 at cV_50Gy_. The correlations between dDVH features were also small relative to those between cDVH features, even for adjacent bins. For example, for dDVH (bin = 8 Gy) features (E), the correlation of 0.58 between dV_5–13Gy_ and dV_13–21Gy_ was small, as was that of 0.44 between dV_5–13Gy_ and dV_45–53Gy_.

Table 2 summarizes the mean (95% confidence interval, CI) AUC resulting from 30 prediction models. The highest AUC of 0.73 (0.72–0.74) was provided by a +dDVH (bin = 4 Gy) model (*p* < 0.05). Another +dDVH model also had a higher AUC than had any cDVH model (*p* < 0.05). In the +dDVH models, for any dose bin, cV60Gy was most frequently selected, as shown in Figure 5. The average number (s.d.) of selected features for 30 prediction models in the cDVH model was 1.0 (0.3), and those in the +dDVH models with bin = 2, 4, 6, and 8 Gy were 5.3 (2.4), 4.8 (1.5), 4.7 (1.3), and 3.7 (1.3), respectively. Figure 6 shows the frequency of the features that provided the highest coefficient in the +dDVH models. As described in Figure 6, the cV60Gy features most frequently selected were not given the highest coefficient in any of the developed +dDVH models, regardless of the number of dose bins.

## 4. Discussion

The prediction of RP has been attempted by many researchers because this side effect can be fatal in some cases. At the same time, increasing numbers of researchers are developing prediction models using machine learning as one of the next-generation strategies for RP prediction [5,6,7]. In this study, we introduce a dDVH into machine learning for RP prediction and demonstrate the predictive performance of the developed model with integrated cDVH and dDVH features.

dDVH features have the advantage of resolving multicollinearity, which often prevents the development of optimal prediction models due to the interdependencies between explanatory variables or the near-linear dependencies between features [9]. In the prediction of RP occurrence, previously reported models have been developed using cDVH features that have a common dose–volume ratio [5,6,7], which causes multicollinearity. For example, between cV_5Gy_ and cV_20Gy_, cV_20Gy_ is fully included in cV_5Gy_. This inclusion means that the determination of the coefficient for V_5Gy_ is partially dependent on that for cV_20Gy_. In contrast, dDVH features prevent this inclusion because the fractional volume of the lung receiving dose is calculated using binned dose. In practice, the Spearman’s coefficient showing the interdependencies between dDVH features was smaller than that for cDVH features (Figure 4 and Figure 5). Thus, dDVH allowed the LASSO to determine the optimal regression coefficients for every feature related to RP occurrence and shrink features that were not related to RP occurrence with the L1 norm.

Further, based on the expressions used in this study, the dDVH features presented the dose–lung relationship in more detail than did cDVH features (Figure 1, Figure 2 and Figure 3). The predictive performance of the developed model was improved by the inclusion of dDVH features, which were able to provide the irradiated volume for each binned dose. The developed model had better predictive power based on the enhancement of its expressive power and the resolution of multicollinearity caused by the use of dDVH features. Among the models developed, that with the best predictive power used dDVH features calculated with bin = 4 Gy. Overall, the calculation of dDVH features with a narrower bottle width led to a more detailed expression of lung irradiation dose. Therefore, models developed using features calculated on a smaller bin size were expected to best predict RP occurrence. In fact, previous studies have predicted the toxicity of radiotherapy using dDVH features with bin sizes of 0.33 [8] and 0.1 Gy [12]. The conflict between this and our results can be partially explained by lung motion caused by respiration [18]. In particular, lung motion cause discrepancies between the dose displayed on the TPS and the dose received by the patient. A dDVH with a narrower bin width tends to cause inconsistency because the volume corresponding to the calculation of each feature is smaller. Using dDVH features, the optimal dose bin in our patient cohort was determined by the LASSO to be 4 Gy, considering the trade-off between expressive power for dDVH features and robustness to lung motion.

With regard to machine learning, in the developed model, the LASSO has the opportunity to demonstrate the relationship between RP occurrence and one or more risk factors using coefficients generated for every risk factor. In model development, the LASSO removes redundant features by shrinking coefficients to zero with a penalizing term expressed in L1 norm. Coefficients for predictive features given greater-than-zero values from LASSO indicate RP occurrence introduced by radiotherapy. Interestingly, our results revealed an infrequent selection of V20Gy and MLD, which have been reported as predictors for RP [1,2,3], regardless of the bin width (Figure 5). However, data previously reported by Hope et al. is consistent with this result, demonstrating that most models include a dosimetric parameter describing the low to middle range of cDVH, such as D35Gy [19]. As far as we know, no paper has previously adapted dDVH for RP prediction. As shown in Figure 5, our results demonstrate that dDVH features were selected without bias toward low-range, mid-range, or high-range doses. In the +dDVH models, the dose bin that determined the highest coefficient in the case of the most frequently selected cV_60Gy_ (Figure 5) was only 8 Gy (Figure 6). It should be noted that the frequency of selection and the magnitudes of the coefficients do not always match.

It is significant that the results for our +dDVH models were obtained using a cohort of patients whose normal lung received doses constrained using cDVH. Our model improves the prediction of RP in patients who received a planned dose to normal lung based on cDVH constraints supplied from the RTOG protocol [13], which is also consistent with previous reports [1,2,3]. Therefore, we recommend the prediction of RP by simultaneous use of cDVH constraints and a prediction model rather than use of a prediction model alone. Second, consideration may be given to differences in the impacts of cDVH constraints on the underlying dose distribution in the lung, which may have subsequent effects on RP occurrence. In addition, the differences in dose constraints may lie in factors that cause unexpected side effects that are not incorporated into our model. In clinics that use cDVH constraints [13] or dosimetric results [1,2,3], it is necessary to develop models using patient cohorts that receive their own cDVH constraints and subsequently predict RP with simultaneous use of cDVH and the model adapted to the clinic.

## 5. Conclusions

In RP prediction models developed based on machine learning, there are often interdependencies between cDVH features that prevent accurate prediction. Our results demonstrate that the inclusion of dDVH features succeeds in improving these interdependencies. The introduction of dDVH features into the machine learning framework has further potential in the development of models for predicting the side effects of radiotherapy.

## Figures and Tables

**Figure 1 diagnostics-12-01354-f001:**
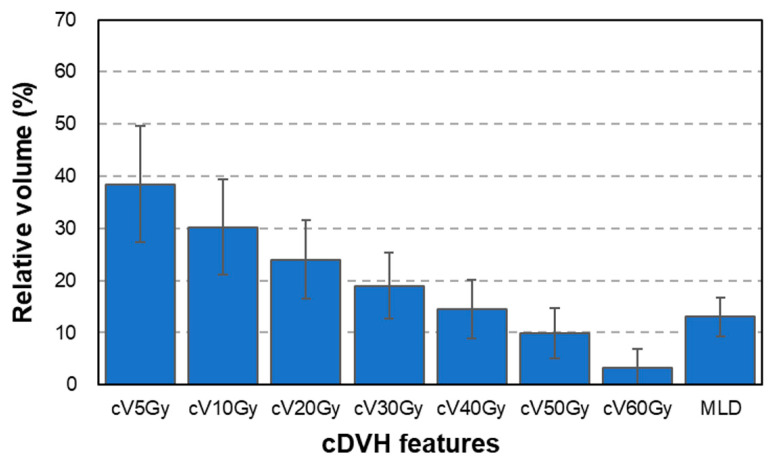
cDVH features of 153 patients in our cohort. Abbreviation: cDVH = cumulative dose–volume histogram. Data presented as mean ± standard deviation.

**Figure 2 diagnostics-12-01354-f002:**
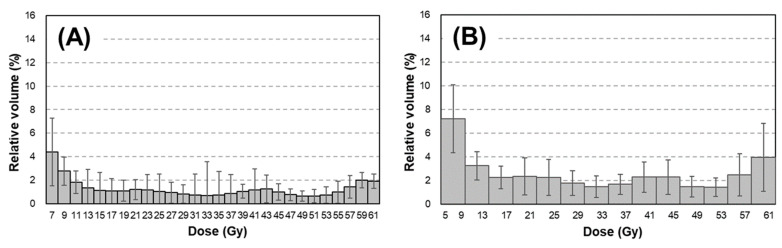
dDVH features calculated with dose bin = 2 Gy (**A**), 4 Gy (**B**), 6 Gy (**C**), and 8 Gy (**D**). Abbreviation: dDVH = differential dose–volume histogram. Data presented as mean ± standard deviation.

**Figure 3 diagnostics-12-01354-f003:**
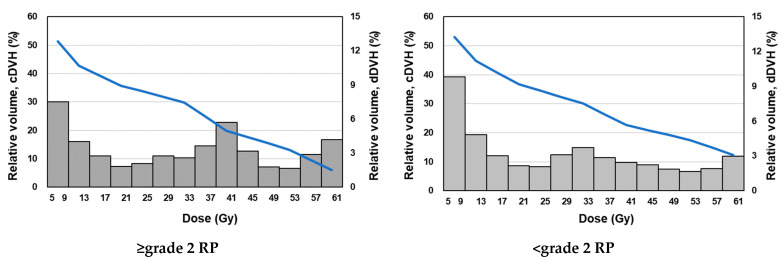
Example of cDVH curve and dDVH features for ≥grade 2 RP and <grade 2 RP on a representative patient. Abbreviations: cDVH = cumulative dose–volume histogram; dDVH = differential dose–volume histogram; RP = radiation pneumonitis.

**Figure 4 diagnostics-12-01354-f004:**
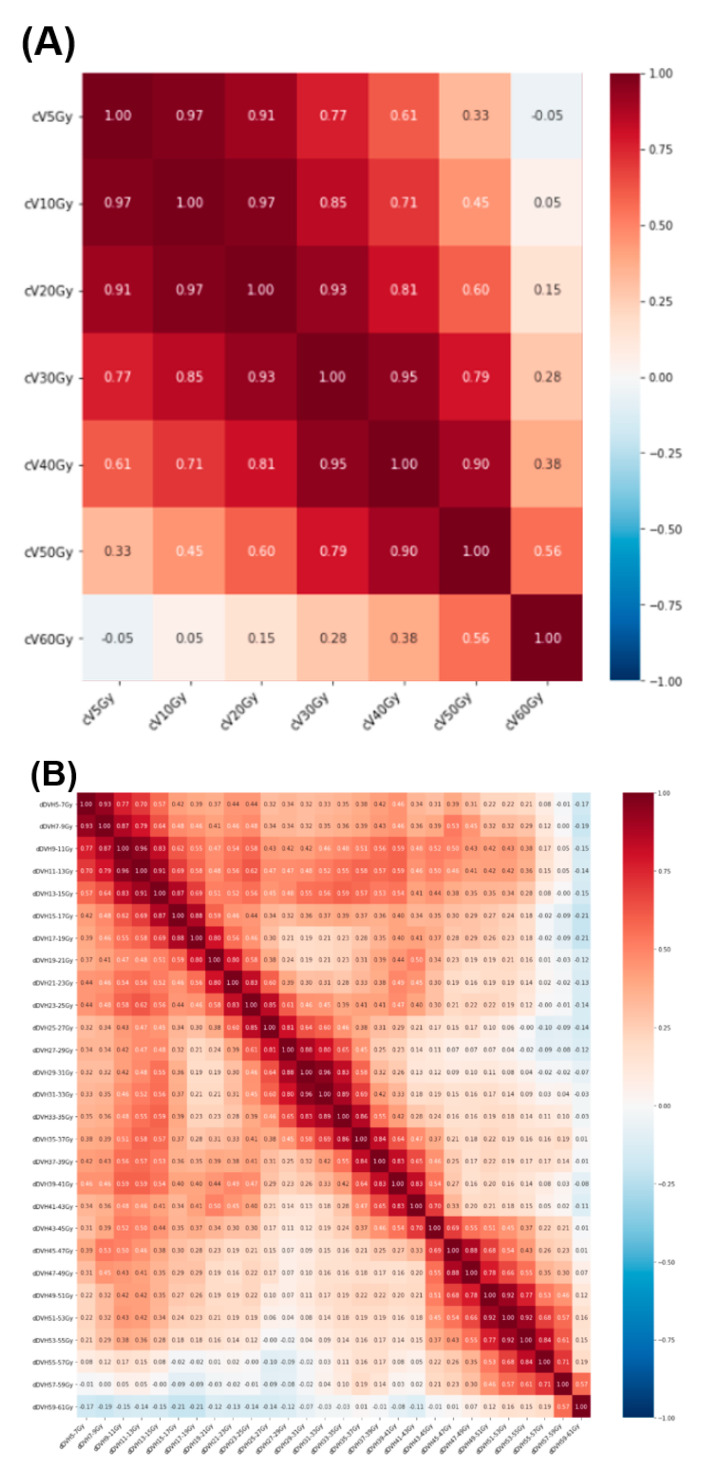
Feature correlation heat map illustrating the correlation between cDVH features (**A**) and dDVH features with dose bin = 2 Gy (**B**), 4 Gy (**C**), 6 Gy (**D**), and 8 Gy (**E**). Abbreviations: cDVH = cumulative dose–volume histogram; dDVH = differential dose–volume histogram.

**Figure 5 diagnostics-12-01354-f005:**
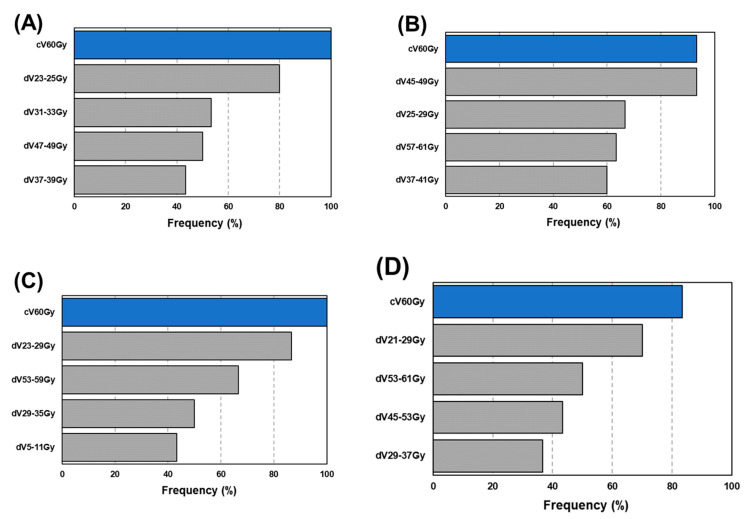
Frequently selected features in the developed model integrating cDVH and dDVH features with dose bin = 2 Gy (**A**), 4 Gy (**B**), 6 Gy (**C**), and 8 Gy (**D**). Abbreviations: cDVH = cumulative dose–volume histogram; dDVH = differential dose–volume histogram.

**Figure 6 diagnostics-12-01354-f006:**
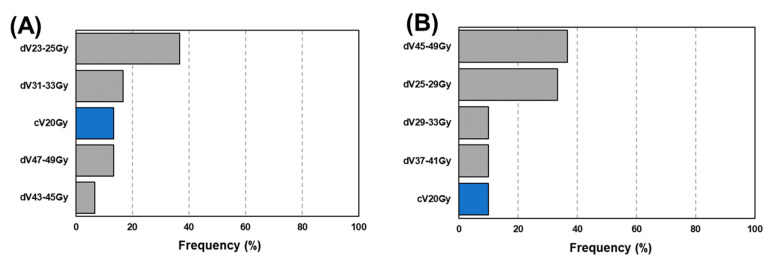
Frequency of features giving the highest coefficient in the developed model integrating cDVH and dDVH features with dose bin = 2 Gy (**A**), 4 Gy (**B**), 6 Gy (**C**), and 8 Gy (**D**). Abbreviations: cDVH = cumulative dose–volume histogram; dDVH = differential dose–volume histogram.

**Table 1 diagnostics-12-01354-t001:** Summary of patient characteristics.

Characteristic		n	%
Gender	Male	130	85.0
	Female	23	15.0
RP classification	<Grade 2	112	73.2
	≥Grade 2	41	26.8
Smoking history	Yes	97	63.4
	No	56	36.6
Total dose	60 Gy	120	78.4
	66 Gy	33	21.6
Dose per fraction	2.0 Gy	152	99.3
	1.8 Gy	1	0.7

Abbreviations: RP = radiation pneumonitis.

**Table 2 diagnostics-12-01354-t002:** Mean and 95% CI interval of the AUC on training and testing data sets for the cDVH model and +dDVH models.

Developed Model	AUC Mean (95% CI)	
	Training Partition	Testing Partition
cDVH	0.62 (0.58–0.65)	0.60 (0.57–0.63)
+dDVH (bin = 2 Gy)	0.71 (0.69–0.74)	0.72 (0.70–0.74)
+dDVH (bin = 4 Gy)	0.73 (0.71–0.76)	0.73 (0.72–0.74)
+dDVH (bin = 6 Gy)	0.69 (0.66–0.71)	0.69 (0.66–0.72)
+dDVH (bin = 8 Gy)	0.68 (0.65–0.72)	0.69 (0.67–0.72)

Abbreviations: AUC = area under curve; CI = confidence interval; cDVH = cumulative dose–volume histogram; dDVH = differential dose–volume histogram.

## Data Availability

Not available.

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
