# Peer review of "Feasibility of Differential Dose—Volume Histogram Features in Multivariate Prediction Model for Radiation Pneumonitis Occurrence"

_diagnostics, 2022, doi:10.3390/diagnostics12061354_

Round 1

Reviewer 2 Report

This an important manuscript. it can be improved with some revision.

  1. The English should be improved.
  2.  The figure must be improved.
  3. The authors should re-write the conclusion to improve the significance of this manuscript.

Reviewer 3 Report

In radiotherapy, care for NSCLC is difficult because healthy lung tissues as well as multiple organs, including the esophagus, trachea, and brachial plexus, are located around the tumor. The unfavorable results of this therapy often contain radiation RP, which is deadly in some cases. Therefore, Katsuta and colleagues proposed a machine learning approach to RP prediction in NSCLC. In addition, this study is the first investigation to use differential dose-volume histogram (dDVH) features in the development of an RP prediction model using machine learning and reveal the model’s predictive interpretation for RP events. Thus, these authors introduce a dDVH into machine learning for RP prediction and establish the predictive execution of the developed model with integrated cDVH and dDVH features.

Comments:

Fantastic work!!

The manuscript is well written.

The methodology is fine and no further control is required.

I found the conclusion to be in line with the evidence and arguments presented, and yes, the authors address the main question beautifully.

The tables are fine.

The figures are of good quality.

However, I have some suggestions that authors should consider that may help to improve the manuscript.

  1. For Figures: See the caption of Fig 2. (Figure 2. dDVH features calculated with dose bin = 2, 4, 6, and 8 Gy.)

It could have been better if the authors had split the figure into A, B, C, and D. It will also be helpful for the readers. Similarly, in Fig. 4, 6, and 7.

  1. Line 36 " Previous studies have". There is a small mistake in the italic font. please fix it.

Round 2

Reviewer 1 Report

Thank you for taking the time to answer my comments. Thank you for your time.

Reviewer 2 Report

It can be accepted.